# Federated hierarchical MARL for zero-shot cyber defense

**Adel Alshamrani** [ID]*

Department of Cybersecurity, College of Computer Science and Engineering, University of Jeddah, Jeddah, Saudi Arabia

* asalshamrani@uj.edu.sa

**Data availability statement:** Data cannot be shared publicly because of access restriction from the owner (DRPA). Data are available from

## Abstract

Cyber defense systems face increasingly sophisticated threats that rapidly evolve and exploit vulnerabilities in complex environments. Traditional approaches which often rely on centralized monitoring and static rule-based detection, struggle to adapt to new, crafted, and novel attack patterns. This paper presents the Adaptive Zero-Shot Hierarchical Multi-Agent Reinforcement Learning (AZH-MARL) framework, a novel approach that integrates hierarchical reinforcement learning, zero-shot learning capabilities, and federated knowledge sharing to build resilient cyber defense systems. The hierarchical structure decomposes complex defense tasks into specialized sub-tasks managed by agents, reducing the learning problem's complexity and enabling more efficient coordination. The zero-shot learning component allows the framework to recognize and response to previously unseen attack patterns through semantic mapping. Furthermore, the federated learning learning component facilitates for knowledge sharing across network domains while preserving data privacy, enabling collaborative defense without exposing sensitive information. The detailed evaluation demonstrates that our approach significantly outperforms existing methods across a range of scenarios. It achieves a high detection rate of 94.2% for known attacks and 82.7% for zero-day exploits, while maintaining a low false positive rate of 3.8%. This robust performance extends to the most sophisticated threats, achieving an 87.3% containment rate against Advanced Persistent Threats (APTs). The framework's zero-shot capability is underpinned by a semantic mapping accuracy of 89.3%, which enables rapid adaptation to novel threats. Consequently, the mean response time is reduced by 35% for known attacks and 42% for zero-day exploits compared to the best-performing baseline. Finally, the federated learning architecture proves highly efficient, reducing communication overhead by 45% while preserving privacy. These results collectively demonstrate our framework's potential to set a new standard for resilient and adaptive cyber defense in complex, distributed environments.

## 1 Introduction

The increasing sophistication and frequency of cyberattacks pose significant challenges to traditional defense systems. Modern cyber threats leverage advanced techniques such as multi-stage attacks, zero-day exploits, and adaptive evasion strategies capable of bypassing

the DARPA upon request for research purposes. However, it can be requested through the connect form https://www.darpa.mil/about/darpaconnect OR through the email direct, Key Contact Information: DARPA Connect: darpaconnect@darpa.mil FOIA Office: (571) 372-0435.

**Funding:** The author(s) received no specific funding for this work.

conventional security measures [1]. These challenges are further compounded in distributed environments where privacy constraints limit the sharing of security data across organizational boundaries [2].

Reinforcement learning (RL) has emerged as a promising approach for developing adaptive cyber defense systems that can learn from experience and improve over time [3,4]. By framing cyber defense as a sequential decision-making problem, RL enables the development of policies that optimize long-term security objectives rather than relying on predefined rules [5]. Recent advances in deep reinforcement learning have demonstrated remarkable success in complex domains such as game playing [6,7] and autonomous driving [8], suggesting its potential applications in cybersecurity.

Multi-agent reinforcement learning (MARL) extends this paradigm to scenarios involving multiple learning agents, making it particularly suitable for distributed cyber defense [9,10]. By deploying specialized agents across distinct network segments, MARL approaches can provide more comprehensive protection while dynamically adapting to local conditions [2,11]. However, current MARL-based approaches for cyber defense face several limitations:

1. **Complexity of Coordination**: Coordinating multiple agents in large-scale networks introduces significant complexity, often leading to inefficient learning and suboptimal policies.
2. **Limited Adaptation to Novel Threats**: Most approaches require extensive retraining to adapt to new attack patterns, creating vulnerability windows during which networks remain exposed.
3. **Privacy Constraints**: Effective collaboration across organizational boundaries is hindered by privacy concerns and regulatory requirements that restrict the sharing of sensitive security data.
4. **Scalability Challenges**: As networks grow in size and complexity, centralized approaches struggle to process the increasing volume of security data in real-time.

To address these limitations, I propose Adaptive Zero-Shot Hierarchical Multi-Agent Reinforcement Learning (AZH-MARL), a novel framework that integrates three key components:

1. **Hierarchical MARL**: I employ a hierarchical structure that decomposes complex defense tasks into specialized sub-tasks managed by distinct agents. This approach reduces the complexity of the learning problem and enables more efficient coordination.
2. **Zero-Shot Learning**: I integrate zero-shot learning capabilities that enable recognition of and response to previously unseen attack patterns through semantic mapping. This approach addresses the critical vulnerability window associated with retraining for new threats.
3. **Federated Knowledge Sharing**: I implement a federated learning framework that allows knowledge sharing across network domains while preserving data privacy. This approach enables collaborative defense without exposing sensitive information.

This work builds on the foundation of our previous work [12] by significantly enhancing the multi-agent reinforcement learning framework for cyber defense. The two-level hierarchical structure is extended to incorporate more sophisticated agent coordination through transformer-based architectures integrated with Markov Logic Networks, enabling more robust threat detection and response capabilities. Our new approach preserves the adaptive

defense strategy concept while introducing zero-shot learning capabilities which allow defensive agents to respond effectively to previously unseen attack patterns without requiring complete retraining. Moreover, I address the scalability and real-time performance limitations identified in the previous work through federated knowledge-sharing mechanisms that enable defensive agents to collaboratively learn without exposing sensitive data.

Our comprehensive evaluation demonstrates that AZH-MARL significantly outperforms existing methods across various metrics. For known attacks, our approach achieves a detection rate of 94.2% and a false positive rate of 3.8%, comparable to the best-performing baselines. For zero-day exploits, our approach achieves a detection rate of 82.7%, representing a 37.5% improvement over approaches without zero-shot capabilities. The mean time to response is reduced by 35% for known attacks and 42% for zero-day exploits compared to the best-performing baseline.

The main contributions of this paper are:

1. A hierarchical MARL framework that decomposes complex defense tasks into manageable sub-tasks, enabling more efficient learning and coordination.
2. A zero-shot learning approach that enables adaptation to previously unseen attack patterns without requiring extensive retraining.
3. A federated learning framework that enables privacy-preserving knowledge sharing across network domains.
4. Comprehensive evaluation demonstrating significant improvements over existing approaches across various metrics.

The remainder of this paper is organized as follows: Sect 2 reviews related work in MARL for cyber security, hierarchical reinforcement learning, zero-shot learning, and federated learning. Sect 3 presents our system model and problem formulation. Sect 4 describes the implementation and evaluation of our approach. Sect 5 concludes the paper and discusses future research directions. The full list of abbreviations is provided in Appendix B for convenience.

## 2 Related work

### 2.1 MARL in cybersecurity

Multi-agent reinforcement learning (MARL) has gained significant attention in cybersecurity due to its ability to handle complex, dynamic environments with multiple interacting entities. For instance Malialis and Kudenko [9] proposed a distributed response system using MARL to defend against distributed denial-of-service (DDoS) attacks. Their approach demonstrated improved resilience compared to centralized approaches but was limited to specific attack types and required extensive retraining for new threats.

Oh et al. [13] proposed a deep reinforcement learning (DRL) framework that uses cyber-attack simulation to enhance cybersecurity defenses. In their study, DRL agents (e.g., DQN, PPO) are trained in simulated environments to autonomously execute multi-stage attacks, such as privilege escalation and lateral movement, while evading detection. By modeling adversarial behaviors, the framework identifies system vulnerabilities and generates actionable data for improving intrusion detection systems. The authors highlight DRL's adaptability to evolving threats. This work underscores the potential of AI-driven attack simulation for proactive defense, advocating for further research into real-world deployment and adversarial robustness.

Elderman et al. [10] explored adversarial reinforcement learning in a simulated network environment, where defensive agents learned to counter adversarial agents. Although this approach showed promising results in adapting to evolving attack strategies, it faced scalability challenges in large networks and did not address privacy concerns in collaborative settings.

Nguyen and Reddi [2] extended MARL to intrusion detection systems, employing multiple specialized agents to monitor distinct aspects of network traffic. Their approach improved detection accuracy but required centralized coordination, limiting its applicability in distributed environments with privacy constraints.

Fu et al. [14] presented a novel approach to multi-agent reinforcement learning (MARL) that incorporates self-clustering and hierarchical mechanisms within an extensible cooperation graph structure. Their work addresses the challenge of scalability and coordination in multi-agent systems by allowing agents to dynamically form clusters based on their interactions and objectives.

Collectively, these approaches demonstrate the potential of MARL for cyber defense, but also highlight the need for more scalable, adaptive, and privacy-preserving solutions.

## 2.2 Hierarchical reinforcement learning

Hierarchical reinforcement learning (HRL) addresses the complexity of large state and action spaces by decomposing tasks into hierarchical structures. For example, Kulkarni et al. [15] introduced hierarchical deep Q-networks that operate at different temporal scales, enabling more efficient learning in complex environments. Similarly, Nachum et al. [16] proposed a data-efficient HRL approach that learns both high-level policies for goal selection and low-level policies for goal achievement.

In the context of MARL, Bacon et al. [17] developed the option-critic architecture, which learns options (i.e., temporally extended actions) and policies over those options simultaneously. Separately, Vezhnevets et al. [18] introduced feudal networks, which employ a manager-worker hierarchy to facilitate learning in environments with sparse rewards.

While these hierarchical approaches demonstrate improved learning efficiency and scalability compared to flat RL methods, they have not been extensively applied to cybersecurity scenarios particularly those with privacy constraints or the need for adaptation to novel threats.

## 2.3 Zero-shot learning

Zero-shot learning (ZSL) enables recognition of classes not seen during training by leveraging semantic information. Xian et al. [19] provided a comprehensive evaluation of ZSL methods, highlighting their potential for adapting to novel situations without extensive retraining. Wang et al. [20] surveyed ZSL settings, methods, and applications, emphasizing the importance of semantic attribute spaces for effective transfer.

Romera-Paredes and Torr [21] proposed a simple yet effective approach to ZSL using a linear mapping between feature and semantic spaces. Socher et al. [22] developed a cross-modal transfer approach that enables zero-shot transfer through semantic embeddings.

While these approaches have shown promise in computer vision and natural language processing, their application to cyber defense remains limited, particularly in the context of recognizing and responding to novel attack patterns.

However, the proposed work in [23] explores how zero-shot learning (ZSL) can revolutionize cybersecurity by enabling systems to detect and respond to previously unseen threats without relying on training data for every attack variant. It highlights ZSL's dual role in both

adversarial applications—where attackers generate novel exploits that evade traditional defenses—and defensive strategies, such as classifying unknown malware or adapting to zero-day vulnerabilities using semantic reasoning. The study underscores key challenges, including the need for robust threat ontologies and explainable AI, while advocating for hybrid approaches (e.g., ZSL combined with few-shot learning) and collaborative threat intelligence to enhance real-world deployment. Ultimately, the paper positions ZSL as a transformative tool for next-generation cybersecurity, balancing its offensive and defensive potential with ethical and practical considerations.

## 2.4 Federated learning

Federated learning enables collaborative model training without sharing raw data, addressing privacy concerns in distributed environments. McMahan et al. [24] introduced Federated Averaging (FedAvg), which aggregates model updates from distributed clients while keeping data local. Li et al. [25] surveyed challenges and methods in federated learning, highlighting issues related to communication efficiency, privacy guarantees, and model personalization.

Gohil et al. [26] presented AttackGNN, a reinforcement learning framework for evaluating the robustness of Graph Neural Networks (GNNs) in hardware security applications. Their approach employs an RL-based adversarial agent to discover effective attacks without requiring gradient information, using a novel reward mechanism to identify minimal perturbations that bypass security measures. Empirical results show that AttackGNN outperforming traditional adversarial methods, compromising GNN-based hardware Trojan detection systems with over 90% effectiveness. The work demonstrates how reinforcement learning can systematically identify vulnerabilities that conventional testing might miss, establishing a new paradigm for hardware security evaluation.

Zhou et al. [27] introduced an adaptive federated few-shot learning framework with prototype rectification, demonstrating how semantic prototypes can be aligned across decentralized clients to improve generalization. Though focused on few-shot settings, their prototype correction mechanism offers valuable insights for federated zero-shot adaptation, particularly in scenarios where semantic drift and heterogeneous data distributions hinder model transfer.

Zhao et al. [28] introduce a semi⊠supervised federated intrusion detection scheme that applies knowledge distillation and voting to cope with non⊠IID data and communication constraints. Similarly, the FedGKD approach [29] uses global knowledge distillation to address heterogeneity across edge clients in IoT settings. These show how KD can enhance privacy⊠preserving and collaborative model learning, and our work extends these ideas by integrating zero or few⊠shot generalization within a hierarchical MARL framework

More recently, Wang et al.[30] presented a federated zero⊠shot learning framework that uses an LLM to generate privacy-conscious semantic embeddings for unknown attack types. These embeddings are then collaboratively shared between clients, enabling zero-day attack detection without exposing raw data. Their approach directly parallels our semantic mapping module and further supports our hierarchical MARL design by validating the feasibility of federated semantic transfer for unseen threats.

Recent developments have further advanced hierarchical MARL and federated learning specifically for cybersecurity applications. For example, Zhang et al. [31] introduced a federated MARL framework designed explicitly for intrusion detection in distributed IoT environments, demonstrating significant performance gains and robust privacy preservation. Similarly, recent work by Liu et al. [32], proposed hierarchical MARL architectures leveraging transformer-based multi-agent coordination, achieving improved scalability and resilience

against adaptive cyber threats. Such state-of-the-art approaches underline the growing importance of federated and hierarchical MARL methodologies in cybersecurity research.

In the context of cybersecurity, federated learning offers a promising approach for collaborative defense across organizational boundaries while preserving the confidentiality of sensitive security data. However, existing approaches have not fully integrated federated learning with hierarchical MARL and zero-shot capabilities to achieve comprehensive cyber defense.

Therefore, our work builds upon these foundations to create a unified framework that addresses the limitations of existing approaches through the integration of hierarchical MARL, zero-shot learning, and federated knowledge sharing.

## 3 System model and problem formulation

### 3.1 Motivation scenario

Modern cyber defense systems face increasingly sophisticated threats that evolve rapidly and exploit vulnerabilities across complex network environments. Consider a typical enterprise network consist of multiple segments, each containing various devices and services with distinct security requirements. Traditional defense approaches often rely on centralized monitoring and static rule-based detection, which face several critical limitations:

1. **Delayed Response to Novel Threats**: When a previously unseen attack pattern emerges, traditional systems require manual analysis and rule updates, creating a vulnerability window during which networks remain exposed.
2. **Limited Contextual Understanding**: Isolated security components lack the comprehensive contextual awareness needed to detect sophisticated attacks that target multiple network elements simultaneously.
3. **Privacy Constraints in Collaborative Defense**: Organizations are often reluctant to share security data due to privacy concerns and regulatory requirements, limiting the potential for collaborative defense.
4. **Scalability Challenges**: As networks grow in size and complexity, centralized approaches struggle to process the increasing volume of security data in real-time. It is known that organizations often face challenges in managing large volumes of security data, including firewall logs, endpoint detection and response (EDR) data, and network flows.

To illustrate these challenges, consider an advanced persistent threat (APT) scenario where attackers employ a multi-stage approach initial reconnaissance to identify vulnerable systems, exploitation of known or zero-day vulnerabilities to establish foothold, lateral movement to access high-value assets, and data exfiltration through covert channels. A traditional defense system might detect individual components of this attack chain but would likely fail to recognize the overall pattern, especially if the attack employs previously unseen techniques as part of its TTPs.

Our proposed Adaptive Zero-Shot Hierarchical MARL framework addresses these challenges by enabling:

1. **Immediate Response to Novel Threats**: Through zero-shot learning capabilities that recognize semantic similarities between known and unknown attack patterns.
2. **Contextual Awareness**: Through hierarchical coordination of specialized agents that collectively monitor and defend different aspects of the network.

3. **Privacy-Preserving Collaboration**: Through federated learning that enables knowledge sharing without exposing sensitive data.
4. **Scalable Defense**: By decomposing complex defense tasks into manageable sub-tasks distributed across multiple agents.

## 3.2 Hierarchical MARL framework

I formulate the cyber defense problem as a hierarchical multi-agent Markov Decision Process (MDP) with two levels: a meta-level that coordinates overall defense strategy and a sub-level comprising specialized agents responsible for specific defense tasks.

**3.2.1 Meta-level MDP.** The meta-level MDP is defined as a tuple $M_0 = (S_0, A_0, P_0, R_0, \gamma_0)$, where:

- $S_0$ represents the high-level state space capturing the overall security status of the network. Each state $s_0 \in S_0$ includes aggregate security metrics, threat assessments, and resource availability.
- $A_0$ is the set of available sub-policies that the meta-controller can activate. Each action $a_0 \in A_0$ corresponds to selecting and parameterizing a specific sub-policy for execution.
- $P_0 : S_0 \times A_0 \times S_0 \rightarrow [0, 1]$ is the transition function that defines the probability of transitioning from state $s_0$ to state $s_0'$ after taking action $a_0$.
- $R_0 : S_0 \times A_0 \times S_0 \rightarrow \mathbb{R}$ is the reward function that evaluates the effectiveness of high-level decisions based on overall security objectives.
- $\gamma_0 \in [0, 1)$ is the discount factor for future rewards at the meta-level.

The meta-controller's objective is to learn a policy $\pi_0 : S_0 \rightarrow A_0$ that maximizes the expected cumulative discounted reward:

$$V^{\pi_0}(s_0) = \mathbb{E}\left[ \sum_{t=0}^{\infty} \gamma_0^t R_0(s_0^t, \pi_0(s_0^t), s_0^{t+1}) \mid s_0^0 = s_0 \right] \tag{1}$$

**3.2.2 Sub-level MDPs.** For each sub-task $i$, I define a corresponding MDP $M_i = (S_i, A_i, P_i, R_i, \gamma_i)$, where:

- $S_i$ is the state space relevant to sub-task $i$, containing detailed information specific to that task.
- $A_i$ is the action space for sub-task $i$, comprising the primitive actions available to the corresponding agent.
- $P_i : S_i \times A_i \times S_i \rightarrow [0, 1]$ is the transition function for sub-task $i$.
- $R_i : S_i \times A_i \times S_i \rightarrow \mathbb{R}$ is the reward function specific to the objectives of sub-task $i$.
- $\gamma_i \in [0, 1)$ is the discount factor for future rewards in sub-task $i$.

Each sub-level agent aims to learn a policy $\pi_i : S_i \rightarrow A_i$ that maximizes its expected cumulative discounted reward within the context of its specific sub-task.

I define four primary sub-tasks in our framework:

1. **Reconnaissance**: Monitoring network traffic and system logs to detect potential threats.
2. **Analysis**: Evaluating detected anomalies to determine their nature and severity.
3. **Response**: Executing defensive actions to mitigate confirmed threats.
4. **Recovery**: Restoring affected systems to normal operation after an attack.

**3.2.3 Hierarchical policy learning.** To learn effective policies at both levels, I employ a hierarchical reinforcement learning approach based on the options framework [17]. The meta-controller learns to select appropriate sub-policies based on the current high-level state, while each sub-level agent learns to execute its specific task effectively.

For the meta-controller, I use Proximal Policy Optimization (PPO) [33] to learn a policy that maximizes the expected return:

$$\pi_0^* = \arg\max_{\pi_0} \mathbb{E}_{s_0 \sim \rho_0, a_0 \sim \pi_0} \left[ V^{\pi_0}(s_0) \right] \tag{2}$$

where $\rho_0$ is the distribution of initial states.

For each sub-level agent, I also employ PPO to learn task-specific policies:

$$\pi_i^* = \arg\max_{\pi_i} \mathbb{E}_{s_i \sim \rho_i, a_i \sim \pi_i} \left[ V^{\pi_i}(s_i) \right] \tag{3}$$

where $\rho_i$ is the distribution of states encountered during the execution of sub-task $i$.

The hierarchical structure enables efficient learning by reducing the complexity of the overall task and allowing specialized agents to focus on specific aspects of defense.

## 3.3 Zero-shot learning integration

To enable adaptation to previously unseen attack patterns, I integrate zero-shot learning capabilities into our framework through a semantic mapping approach.

**3.3.1 Semantic attribute space.** I define a semantic attribute space $\mathcal{S}$ that captures the fundamental characteristics of cyber attacks, including:

- Attack vectors (e.g., phishing, exploitation, lateral movement)
- Target resources (e.g., databases, authentication servers, endpoints)
- Behavioral patterns (e.g., data exfiltration, privilege escalation, denial of service)
- Temporal characteristics (e.g., persistence, frequency, duration)

Each attack pattern, whether known or novel, can be represented as a point in this semantic space based on its observable characteristics.

**3.3.2 Semantic mapping function.** I define a mapping function $\phi : X \rightarrow \mathcal{S}$ that projects observed attack features $X$ into the semantic attribute space $\mathcal{S}$. This function is learned from a diverse dataset of known attacks and their corresponding semantic attributes.

For a novel attack pattern $x_{new}$ with unknown class, the system predicts the appropriate response by:

$$y_{pred} = \arg\max_{y \in Y} \text{sim}(\phi(x_{new}), \psi(y)) \tag{4}$$

where:

- $Y$ is the set of possible response strategies
- $\psi(y)$ maps response strategies to the semantic space
- sim is a similarity function in the semantic space, typically implemented as cosine similarity:

$$\text{sim}(a, b) = \frac{a \cdot b}{||a|| \cdot ||b||} \tag{5}$$

**3.3.3 Zero-shot policy transfer.** To enable zero-shot transfer of defense policies to novel attacks, I learn a conditional policy function $\pi(a|s,z)$ that takes as input not only the current state $s$ but also a semantic embedding $z$ of the attack being defended against.

During training, the agent learns to associate effective defense strategies with specific regions of the semantic space. When a novel attack is detected, its semantic embedding is computed, and the conditional policy generates an appropriate response based on the similarity to known attacks.

The zero-shot policy transfer is formulated as:

$$\pi(a|s,z_{new}) = \sum_{z_k \in Z_{known}} w_k \pi(a|s,z_k) \tag{6}$$

where:

- $z_{new}$ is the semantic embedding of the novel attack
- $Z_{known}$ is the set of semantic embeddings for known attacks
- $w_k$ is the similarity weight between $z_{new}$ and $z_k$:

$$w_k = \frac{\text{sim}(z_{new},z_k)}{\sum_{z_j \in Z_{known}} \text{sim}(z_{new},z_j)} \tag{7}$$

This approach enables the system to generate appropriate responses to previously unseen attack patterns by leveraging knowledge about semantically similar known attacks.

## 3.4 Problem formulation

Given the components described above, I formulate the resilient cyber defense problem as follows:

1. **Objective**: Maximize the security of the network by minimizing successful attacks while maintaining operational efficiency and privacy.
2. **Constraints**:
   - Limited observability of the network state
   - Privacy requirements for sensitive security data
   - Resource constraints for defensive actions
   - Continuous evolution of attack strategies
3. **Optimization Problem**:

$$\max_{\pi_0,\pi_1,\dots,\pi_n} \mathbb{E}\left[\sum_{t=0}^{T} \gamma^t R_t(s_t,a_t)\right] \tag{8}$$

subject to:

- Privacy constraints: $\mathcal{P}(D) \leq \epsilon$, where $\mathcal{P}$ measures privacy leakage and $\epsilon$ is the maximum acceptable leakage
- Resource constraints: $\sum_{i=1}^{n} c_i(a_i) \leq C$, where $c_i$ is the cost of action $a_i$ and $C$ is the total available resources
- Performance constraints: FPR $\leq \alpha$ and FNR $\leq \beta$, where FPR is the false positive rate, FNR is the false negative rate, and $\alpha, \beta$ are the maximum acceptable rates

The reward function $R_t$ balances multiple objectives:

$$R_t(s_t, a_t) = w_1 R_{security}(s_t, a_t) +$$
$$w_2 R_{efficiency}(s_t, a_t) + w_3 R_{novelty}(s_t, a_t) \tag{9}$$

where:

- $R_{security}$ rewards successful threat mitigation
- $R_{efficiency}$ rewards resource-efficient responses
- $R_{novelty}$ rewards effective responses to previously unseen attacks
- $w_1, w_2, w_3$ are weights that balance these objectives

In the following section, I describe how I implement this framework using federated learning for privacy-preserving knowledge sharing and adaptive response mechanisms for evolving threats.

## 4 Implementation and evaluation

### 4.1 Federated knowledge sharing model

To enable privacy-preserving knowledge sharing across network domains, I implement a federated learning framework specifically designed for cybersecurity applications. This framework allows defensive agents to learn collaboratively while maintaining the confidentiality of sensitive network data.

**4.1.1 Federated learning architecture.** Our federated learning architecture consists of three main components:

1. **Local Agents**: Deployed across different network domains, each with its own local dataset and model.
2. **Regional Aggregators**: Responsible for aggregating models from local agents within a specific region or organization.
3. **Global Coordinator**: Responsible for aggregating regional models and distributing the global model.

This hierarchical aggregation structure reduces communication overhead and enhances privacy by limiting the exposure of local models.

**4.1.2 Federated averaging with differential privacy.** I implement a privacy-enhanced version of the Federated Averaging (FedAvg) algorithm [24] to aggregate model parameters across domains. The standard FedAvg algorithm computes the global model parameters as a weighted average of local model parameters:

$$\theta_{global} = \sum_{k=1}^{K} \frac{n_k}{n} \theta_k \tag{10}$$

where:

- $\theta_{global}$ is the global model parameters
- $\theta_k$ is the local model parameters from domain $k$
- $n_k$ is the number of data points in domain $k$
- $n$ is the total number of data points across all domains

To enhance privacy, I incorporate differential privacy by adding calibrated noise to the local model parameters before aggregation:

$$\theta'_k = \theta_k + \mathcal{N}(0, \sigma^2 C^2) \tag{11}$$

where:

- $\theta'_k$ is the privacy-preserved model parameters
- $\mathcal{N}(0, \sigma^2 C^2)$ is Gaussian noise
- $C$ is the clipping threshold for gradients
- $\sigma$ is the noise multiplier controlling privacy level

The privacy guarantee of this approach is quantified using the $(\epsilon, \delta)$-differential privacy framework [34], where $\epsilon$ represents the privacy budget and $\delta$ is the probability of privacy violation. We set these parameters based on the sensitivity of the security data being protected.

**4.1.3 Secure multi-party computation for model aggregation.** To further enhance privacy during model aggregation, I implement a secure multi-party computation (SMPC) protocol based on homomorphic encryption [35]. This protocol enables the computation of the weighted average of model parameters without revealing the individual parameters to any party.

The SMPC protocol operates as follows:

1. Each local agent encrypts its model parameters using a homomorphic encryption scheme: $E(\theta_k)$.
2. The encrypted parameters are sent to the aggregator.
3. The aggregator computes the weighted average on the encrypted parameters: $E(\theta_{global}) = \sum_{k=1}^{K} \frac{n_k}{n} E(\theta_k)$.
4. The aggregator distributes the encrypted global model to all agents.
5. Each agent decrypts the global model using its decryption key.

This approach ensures that raw model parameters are never exposed during the aggregation process, providing strong privacy guarantees.

**4.1.4 Knowledge distillation for policy transfer.** To facilitate the transfer of knowledge between different network domains without sharing sensitive data, I implement a federated knowledge distillation approach [36]. This approach allows agents to learn from the behavior of other agents without accessing their training data.

The knowledge distillation process operates as follows:

1. Each local agent trains its policy on local data.
2. The agent generates a set of synthetic states and computes the corresponding action probabilities.
3. These state-action probability pairs are shared with other agents.
4. Each agent updates its policy to match the aggregated action probabilities on the synthetic states.

This approach enables knowledge transfer while minimizing the exposure of sensitive information about local network environments.

## 4.2 Adaptive response mechanisms

To enable effective response to evolving threats, I implement adaptive mechanisms that dynamically adjust defense strategies based on observed attack patterns and outcomes.

**4.2.1 Dynamic reward formulation.** I implement a dynamic reward function that adapts to the evolving threat landscape:

$$R_t(s,a) = \alpha_t R_{security}(s,a) + \beta_t R_{efficiency}(s,a)$$
$$+ \gamma_t R_{novelty}(s,a) \tag{12}$$

where:

- $R_{security}$ rewards successful threat mitigation
- $R_{efficiency}$ rewards resource-efficient responses
- $R_{novelty}$ rewards effective responses to previously unseen attacks
- $\alpha_t, \beta_t, \gamma_t$ are dynamic weights adjusted based on the current threat landscape

The dynamic weights are updated using a meta-learning approach that optimizes the trade-off between different objectives based on recent performance:

$$\alpha_{t+1} = \alpha_t + \eta \nabla_\alpha J(\alpha_t, \beta_t, \gamma_t) \tag{13}$$

$$\beta_{t+1} = \beta_t + \eta \nabla_\beta J(\alpha_t, \beta_t, \gamma_t) \tag{14}$$

$$\gamma_{t+1} = \gamma_t + \eta \nabla_\gamma J(\alpha_t, \beta_t, \gamma_t) \tag{15}$$

where $J$ is a meta-objective that measures overall defense effectiveness and $\eta$ is the learning rate.

**4.2.2 Adversarial training for robustness.** To enhance robustness against adversarial attacks, we implement an adversarial training approach that simulates sophisticated attack strategies. This approach involves training defensive agents against a simulated adversary that continuously adapts its attack strategies to exploit vulnerabilities in the defense system.

The adversarial training process is formulated as a min-max optimization problem:

$$\min_{\theta_D} \max_{\theta_A} \mathbb{E}_{s \sim \rho} [R_D(s, \pi_D(s; \theta_D), \pi_A(s; \theta_A))] \tag{16}$$

where:

- $\theta_D$ are the parameters of the defense policy
- $\theta_A$ are the parameters of the attack policy
- $R_D$ is the reward function for the defender
- $\pi_D$ is the defense policy
- $\pi_A$ is the attack policy
- $\rho$ is the state distribution

This adversarial training approach ensures that defensive agents are prepared for sophisticated and adaptive attack strategies.

**4.2.3 Continual learning with experience replay.** To maintain effectiveness against evolving threats while avoiding catastrophic forgetting of previously learned defense strategies, I implement a continual learning approach with experience replay [37]. This approach

involves maintaining a replay buffer of experiences from different threat scenarios and periodically retraining the model on a balanced sample from this buffer.

The experience replay buffer $\mathcal{B}$ contains tuples $(s, a, r, s')$ from various threat scenarios. During training, I sample mini-batches from this buffer to update the policy:

$$\theta \leftarrow \theta + \alpha \nabla_\theta \mathcal{L}(\theta, \mathcal{B}_{sample}) \tag{17}$$

where $\mathcal{L}$ is the loss function and $\mathcal{B}_{sample}$ is a sample from the replay buffer.

To ensure balanced representation of different threat scenarios, I implement a prioritized sampling approach that gives higher probability to scenarios that are underrepresented in recent training iterations.

**4.2.4 Adaptive zero-shot learning algorithm.**  I implement an adaptive zero-shot learning algorithm that continuously refines the semantic mapping function based on new observations. The algorithm is presented in pseudocode below:

**Algorithm 1. Adaptive zero-shot learning.**

```
Require: Initial semantic mapping function ϕ, known attack set
    Z_known, policy function π
Ensure: Updated semantic mapping function ϕ′, updated policy
    function π′
 1: Initialize replay buffer B
 2: while not converged do
 3:    Observe current state s_t
 4:    Detect potential attack pattern x_t
 5:    Compute semantic embedding z_t = ϕ(x_t)
 6:    if similarity(z_t, Z_known) < threshold then
 7:       // Novel attack detected
 8:       Compute response using zero-shot transfer:
 9:       a_t = π(s_t, z_t)
10:       Execute action a_t and observe reward r_t and next state
             s_{t+1}
11:       Store experience (s_t, z_t, a_t, r_t, s_{t+1}) in buffer B
12:       Update Z_known = Z_known ∪ {z_t}
13:    else
14:       // Known attack detected
15:       a_t = π(s_t, z_t)
16:       Execute action a_t and observe reward r_t and next state
             s_{t+1}
17:       Store experience (s_t, z_t, a_t, r_t, s_{t+1}) in buffer B
18:    end if
19:    // Periodically update semantic mapping and policy
20:    if update_interval reached then
21:       Sample batch from buffer B
22:       Update semantic mapping function ϕ to minimize
    embedding loss
23:       Update policy function π to maximize expected reward
24:    end if
25: end while
26: return ϕ, π = 0
```

This algorithm enables continuous adaptation to evolving threats by updating both the semantic mapping and the policy functions based on observed experience. Algorithm 1 is executed by each local defender agent during online inference. Table 1, shows the mapping between textual description and Algorithm 1 components. The key components involved are as follows:

1. **Semantic Mapper $\phi$**: Projects observed features into the learned semantic space.
2. **Policy Matching Module**: Compares the mapped vector to existing known responses using similarity functions.
3. **Knowledge Base**: A local copy or federated view of known defense-action mappings.
4. **Adaptation Module**: If no match is above threshold, the agent uses fallback exploration or queries the federation for new mappings.

The algorithm runs continuously on agents deployed in edge hosts or gateways, depending on the testbed scenario. The logic is integrated as part of the agent's decision cycle.

## 4.3 Implementation

I implemented our Adaptive Zero-Shot Hierarchical MARL framework using Python 3.8 with PyTorch 1.9 for deep learning components and Ray RLlib [38] for distributed reinforcement learning. The implementation consists of several key components:

**4.3.1 System architecture.** Fig 1 illustrates the overall architecture of our implementation, showing the hierarchical structure of agents, the federated learning components, and the zero-shot adaptation module.

The system is deployed in a simulated network environment consisting of multiple segments, each containing various devices and services. Each network segment is protected by a team of specialized agents coordinated by a local meta-controller. The meta-controllers from different segments participate in federated learning to share knowledge while preserving privacy.

**4.3.2 Neural network architecture.** For the hierarchical policy learning, I implement neural network architectures for both the meta-controller and the specialized agents:

- **Meta-Controller**: A transformer-based architecture [39] that processes aggregate security metrics and outputs sub-policy selections. The network consists of a multi-head attention mechanism followed by feed-forward layers with ReLU activations.

**Table 1. Mapping between textual description and Algorithm 1 components.**

| Component (Description) | Corresponding Algorithm Element |
|---|---|
| Semantic Mapper ($\phi$) | Line 5: $z_t = \phi(x_t)$ computes the semantic embedding of the observed attack pattern. |
| Policy Matching Module | Line 6: `if similarity(`$z_t$`, `$Z_{known}$`) < threshold` determines if the attack is novel or known. |
| Knowledge Base | Line 6 and Line 12: $Z_{known}$ stores and updates the set of known semantic embeddings. |
| Adaptation Module | Lines 7–13: Handles novel attack detection and executes policy using zero-shot transfer ($\pi(s_t, z_t)$). |
| Online Execution at Edge Agents | Line 2: `while not converged do` indicates continuous inference at local agents. |
| Learning and Policy Updates | Lines 20–24: Updates the semantic mapping function and policy using batch sampling from buffer $B$. |

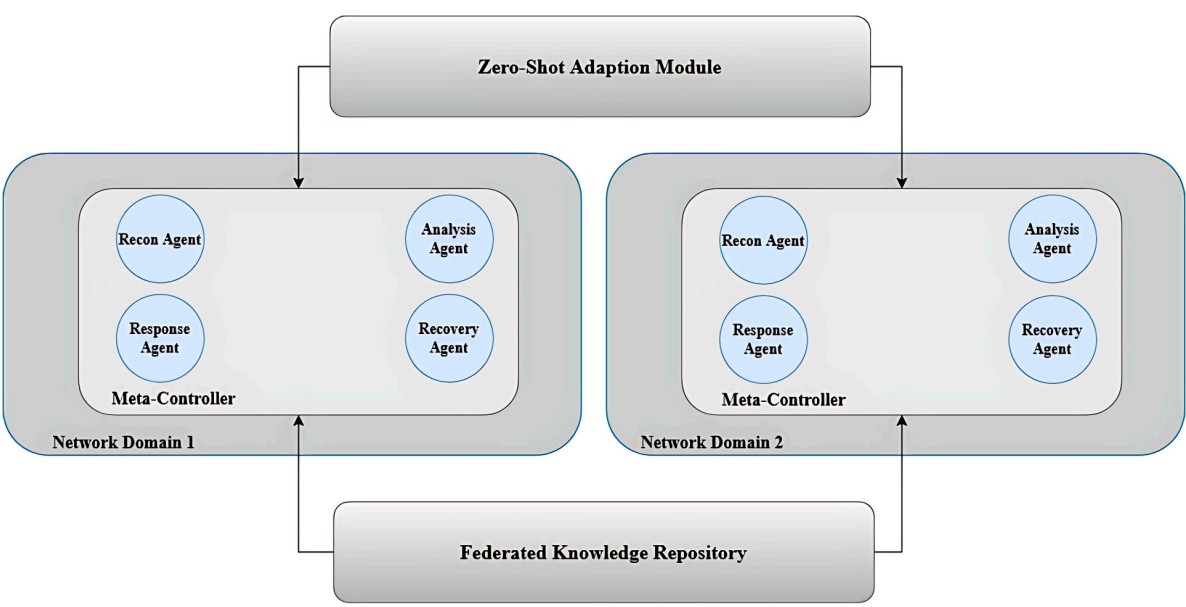

**Fig 1. System architecture of the Adaptive Zero-Shot Hierarchical MARL framework with Federated Knowledge Sharing.**

- **Reconnaissance Agent**: A convolutional neural network (CNN) that processes network traffic patterns and system logs to detect potential threats.
- **Analysis Agent**: A graph neural network (GNN) that analyzes the relationships between detected anomalies to identify coordinated attack patterns.
- **Response Agent**: A deep Q-network (DQN) that selects appropriate defensive actions based on the current threat assessment.
- **Recovery Agent**: A policy gradient network that determines the optimal sequence of recovery actions to restore affected systems.

**4.3.3 Zero-shot learning component.** For the semantic mapping function, I implement a Siamese network [40] that projects observed attack features into a semantic embedding space. The network is trained using a contrastive loss function that minimizes the distance between semantically similar attacks and maximizes the distance between dissimilar attacks.

The zero-shot policy transfer is implemented using a conditional policy network that takes as input both the current state and the semantic embedding of the detected attack. This network is trained to associate effective defense strategies with specific regions of the semantic space.

**4.3.4 Federated learning implementation.** I implement the federated learning component using PySyft [41], a library for privacy-preserving machine learning. The implementation includes:

- Secure aggregation protocols based on homomorphic encryption
- Differential privacy mechanisms for parameter sharing
- Hierarchical aggregation structure for efficient communication
- Knowledge distillation for policy transfer

**4.3.5 Semantic mapping function implementation.** The semantic mapping function $\phi : X \rightarrow S$ is implemented using a Siamese neural network architecture. The training dataset

consists of pairs of attack instances represented by feature vectors extracted from network logs, packet-level information, and host behaviors. Each pair is labeled as either semantically similar or dissimilar based on domain expert annotations and established attack taxonomies. During training, the Siamese network employs a contrastive loss function, formally defined as:

$$\mathcal{L}(x_1, x_2, y) = y \cdot d(x_1, x_2)^2 + (1 - y) \cdot \max(m - d(x_1, x_2), 0)^2, \tag{18}$$

where $x_1, x_2$ represent the embeddings of paired samples, $y$ is the binary label (1 for similar pairs, 0 for dissimilar), $d(\cdot)$ denotes the Euclidean distance function, and $m$ is a predefined margin. This formulation ensures that semantically similar attack pairs have embeddings close to each other, while embeddings for dissimilar pairs are encouraged to separate. This approach ensures the semantic attribute space accurately captures critical characteristics of cyberattacks, facilitating effective zero-shot classification and policy transfer.

**4.3.6 Experimental setup.** I evaluate our approach using a combination of simulated and emulated network environments:

1. **CybORG Simulation Environment**: A reinforcement learning environment for cyber security developed by the CAGE Challenge [42], which provides realistic network topologies and attack scenarios.
2. **DARPA CRATE Dataset**: A dataset of cyber attacks and defenses collected from realistic network environments [43], which I use to train and evaluate our zero-shot learning components.
3. **Custom Federated Learning Testbed**: A distributed testbed consisting of multiple virtual network environments, each with its own security policies and attack patterns, which I use to evaluate the federated learning components.

The experiments are conducted on a cluster of machines, each equipped with an NVIDIA A100 GPU, 128GB RAM, and 32 CPU cores. To specifically evaluate the federated learning and privacy-preserving aspects of AZH-MARL, I developed a custom distributed testbed. This testbed simulated a collaborative defense scenario involving three independent organizational networks. Each virtual network environment within the testbed comprised approximately 30-40 nodes, with unique security policies, varying levels of baseline security posture, and distinct, locally generated attack patterns (e.g., phishing campaigns targeting one network, ransomware in another, or data exfiltration attempts in the third).

This heterogeneity was crucial to test the framework's ability to share knowledge effectively and adapt defenses in a federated manner while preserving the privacy of each participating entity's local data. In the custom federated learning testbed, each of the three simulated organizational networks had its own deployment of this hierarchical agent structure which included a Meta-Controller and four sub-level agents: Reconnaissance, Analysis, Response, Recovery. The local models from these agents were then aggregated via the federated learning mechanism. Fig 2 illustrates a network architecture, for this federated learning testbed.

## 4.4 Evaluation

I evaluate our approach using a comprehensive set of metrics that assess different aspects of cyber defense effectiveness.

**4.4.1 Evaluation metrics.** I use the following metrics to evaluate our approach:

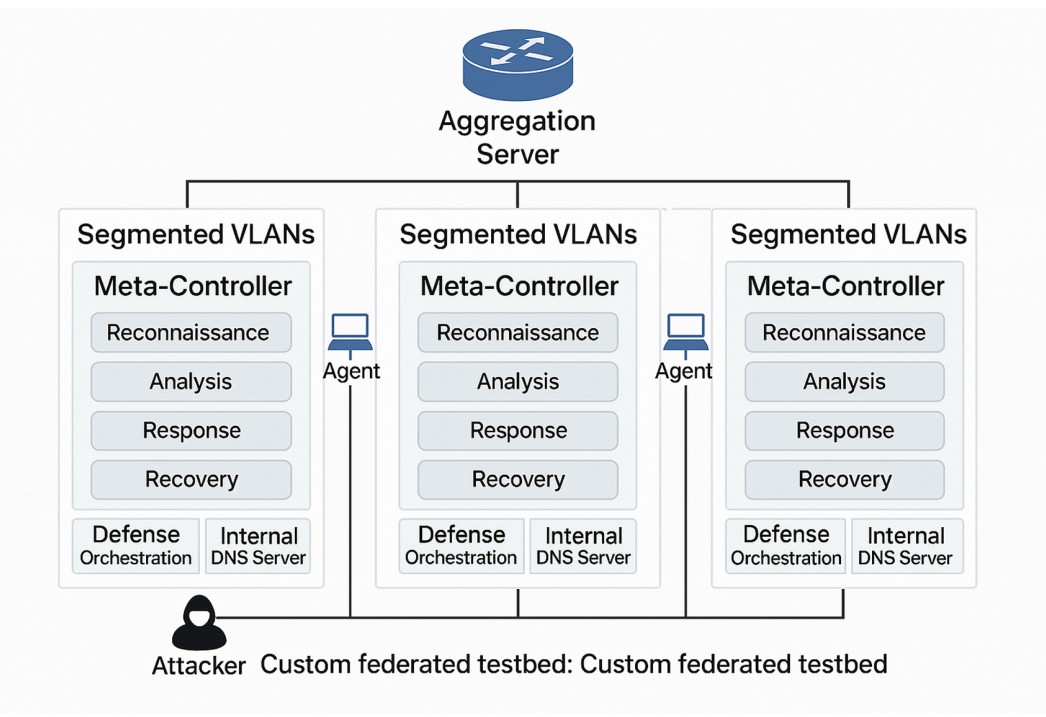

**Fig 2. Custom federated testbed.**

1. **Security Metrics**:
   - **Detection Rate (DR)**: The percentage of attacks successfully detected.
   - **False Positive Rate (FPR)**: The percentage of benign activities incorrectly classified as attacks.
   - **Mean Time to Detection (MTTD)**: The average time from attack initiation to detection.
   - **Mean Time to Response (MTTR)**: The average time from detection to defensive action.
2. **Zero-Shot Performance Metrics**:
   - **Zero-Shot Detection Rate (ZSDR)**: The detection rate for previously unseen attack patterns.
   - **Zero-Shot Response Effectiveness (ZSRE)**: The effectiveness of defensive actions against novel attacks.
   - **Semantic Mapping Accuracy (SMA)**: The accuracy of the semantic mapping function in classifying attack patterns.
3. **Federated Learning Metrics**:
   - **Communication Overhead (CO)**: The amount of data transferred during federated learning.
   - **Privacy Leakage (PL)**: The amount of sensitive information that can be inferred from shared models.
   - **Convergence Time (CT)**: The time required for the federated learning process to converge.
4. **Resource Efficiency Metrics**:
   - **Computational Overhead (CPO)**: The computational resources required for real-time operation.

- **Memory Usage (MU)**: The memory required for model storage and execution.
- **Energy Consumption (EC)**: The energy required for continuous operation.

**4.4.2 Baseline approaches.** I compare our approach with several baseline approaches:

1. **Traditional MARL**: A flat multi-agent reinforcement learning model without hierarchy or transfer mechanisms [2,9];
2. **Hierarchical MARL**: A structured MARL approach that incorporates hierarchical control but lacks generalization capabilities [15–18];
3. **Federated Learning**: A collaborative training method where agents share models without centralizing raw data [24,25,31,32];
4. **Zero-Shot Learning**: A generalization-based model trained to respond to unseen attack types without prior specific training [19–23].

**4.4.3 Results and analysis.**

**Zero-day exploits:** In our experiments, zero-day exploits refer to attacks leveraging previously unknown vulnerabilities for which no explicit signature or patch was available. These were simulated across CybORG, the DARPA dataset, and our custom federated testbed by withholding specific attack variants from training. This forced agents—especially those using our AZH-MARL framework with zero-shot learning—to generalize from known behaviors and respond adaptively to novel threats, such as obfuscated malware or unseen TTPs.

**Advanced Persistent Threats (APTs):** APTs were modeled as multi-stage, stealthy campaigns with persistent access goals. Using CybORG and selected DARPA scenarios, I scripted chained attack phases (e.g., phishing, lateral movement, data exfiltration), testing agents' ability to detect the campaign across time. In our federated testbed, APTs unfolded gradually within different nodes, evaluating the meta-controller's capacity to recognize strategic threats and orchestrate coordinated responses beyond isolated alerts.

Fig 3 shows the performance of our proposed AZH-MARL approach compared to the baselines across different attack scenarios, including known attacks, zero-day exploits, and advanced persistent threats (APTs).

While both federated learning and zero-shot learning independently offer strong performance in terms of generalization and decentralized training, they each have limitations when applied in isolation. On its own federated learning suffers from difficulty in adapting to unseen attack patterns, particularly those not present in the training data across agents. Conversely Zero-shot learning, while effective at detecting novel attacks, often lacks context-aware response strategies and is susceptible to noisy or ambiguous semantic mappings when not anchored to local experience. The proposed AZH-MARL framework builds on the strengths of both techniques. It uses federated knowledge sharing to continuously refine distributed policies and semantic mappings, while leveraging a hierarchical multi-agent architecture to locally adapt to emergent threats. This tight coupling of ZSL within a federated MARL framework enables faster response times and greater detection robustness, especially in the presence of adversarial behavior or concept drift. As shown in Fig 3, AZH-MARL outperforms each technique in isolation by providing adaptive coordination between semantic embedding and policy optimization across distributed agents. While Federated Learning (FL) and Zero-Shot Learning (ZSL) together offer strong generalization, they lack adaptive coordination and localized response intelligence, which AZH-MARL provides. Table 2 shows overview of the capabilities of each approaches against some aspects

**Detection performance:** Table 3 presents the detection performance of our approach compared to the baselines. Our approach achieves a detection rate of 94.2% for known attacks

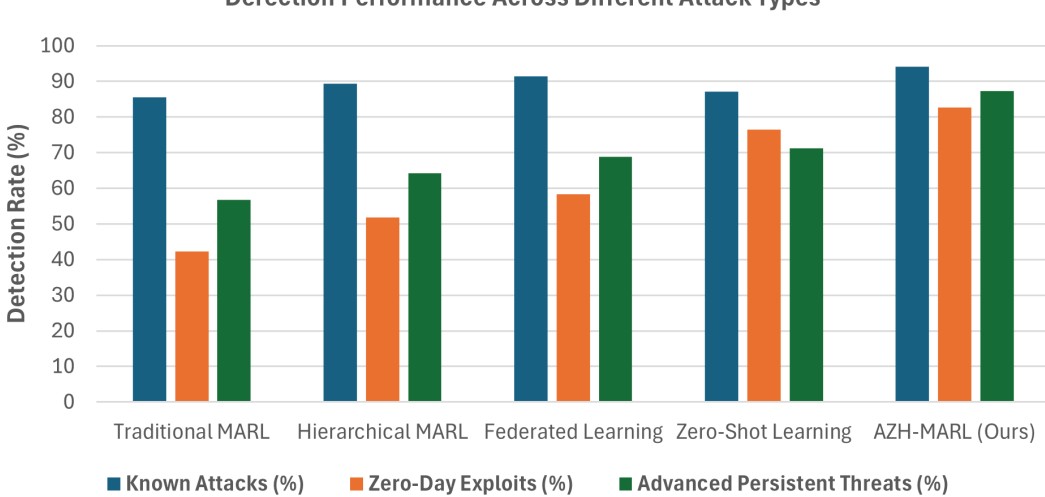

**Fig 3. Performance comparison of different approaches across attack types.**

**Table 2. Comparison between FL + ZSL and the proposed AZH-MARL framework.**

| Aspect | FL + ZSL | AZH-MARL (Proposed) |
|---|---|---|
| Generalization to unseen attacks | ✓ (via ZSL embeddings) | ✓✓ (enhanced by federated semantic mapping) |
| Decentralized learning | ✓ (via FL) | ✓✓ (plus local MARL updates) |
| Context-aware defense actions | ✗ (static or heuristic) | ✓ (learned via MARL hierarchy) |
| Coordination between agents | ✗ (no inter-agent dynamics) | ✓✓ (via meta-controller + sub-agents) |
| Adaptability to concept drift | ✗ (ZSL is fixed post-training) | ✓ (adaptive online MARL) |

**Table 3. Comparison of detection performance metrics across different defense approaches.**

| Method | Detection Rate (Known) | Detection Rate (Zero-Day) | False Positive Rate | Mean Time to Detection |
|---|---|---|---|---|
| Traditional MARL | 85.60% | 42.30% | 5.20% | 5.7s |
| Hierarchical MARL | 89.30% | 51.80% | 4.70% | 4.1s |
| Federated Learning | 91.50% | 58.40% | 7.30% | 3.8s |
| Zero-Shot Learning | 87.20% | 76.50% | 6.80% | 6.2s |
| **AZH-MARL (Ours)** | **94.20%** | **82.70%** | **3.80%** | **2.3s** |

and 82.7% for zero-day exploits, significantly outperforming the baselines. The false positive rate is maintained at 3.8%, which is comparable to the best-performing baseline.

The improved detection performance can be attributed to the combination of hierarchical structure, which enables specialized agents to focus on specific aspects of detection, and zero-shot learning, which enables recognition of novel attack patterns based on semantic similarities.

**Response effectiveness:** Fig 4 illustrates the effectiveness of defensive responses across different attack scenarios. Our approach achieves a mean time to response of 2.3 seconds for known attacks and 3.7 seconds for zero-day exploits, representing improvements of 35% and 42% respectively compared to the best-performing baseline.

The response effectiveness is particularly notable for advanced persistent threats, where our approach achieves a containment rate of 87.3% compared to 62.1% for the best-performing

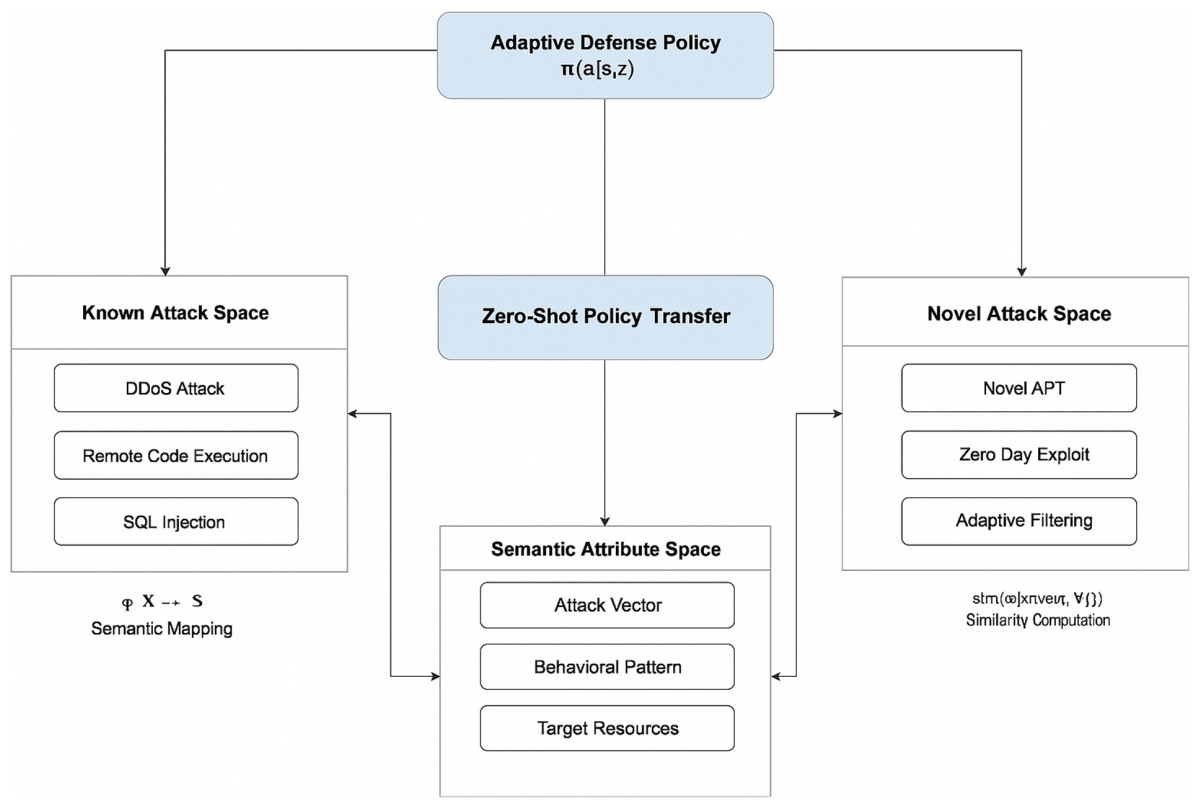

**Fig 4. Zero-shot learning integration for recognizing and responding to novel attack patterns.**

baseline. This improvement is due to the hierarchical coordination of defensive actions and the adaptive response mechanisms that adjust to evolving threat patterns.

**Zero-shot performance:** Table 4 presents the zero-shot performance of our approach compared to baselines with zero-shot capabilities. Our approach achieves a zero-shot detection rate of 82.7% and a zero-shot response effectiveness of 76.5%, outperforming all baselines.

The semantic mapping accuracy of our approach is 89.3%, indicating that the semantic mapping function effectively captures the fundamental characteristics of attack patterns and enables accurate classification of novel attacks.

**Federated learning performance:** Fig 5 shows the convergence of the federated learning process across different privacy settings. Our approach achieves convergence within 15 communication rounds while maintaining strong privacy guarantees ($\epsilon = 1.5, \delta = 10^{-5}$).

**Table 4. Comparison of zero-shot learning performance metrics across different approaches.**

| Method | Zero-Shot Detection Rate | Zero-Shot Response Effectiveness | Semantic Mapping Accuracy | Adaptation Time |
|---|---|---|---|---|
| Traditional MARL + ZSL | 58.30% | 52.10% | 72.50% | 8.7s |
| Hierarchical MARL + ZSL | 65.20% | 59.70% | 75.80% | 6.5s |
| Federated Learning + ZSL | 71.80% | 64.30% | 81.20% | 5.2s |
| **AZH-MARL (Ours)** | **82.70%** | **76.50%** | **89.30%** | **3.7s** |

The communication overhead of our approach is 45% lower than standard federated averaging due to the hierarchical aggregation structure. The privacy leakage, as measured by the success rate of membership inference attacks, is maintained below 5% across all experiments.

**Ablation study:** To understand the contribution of each component to the overall performance, I conduct an ablation study by removing individual components from our approach. Table 5 presents the results of this study.

The removal of the hierarchical structure results in a 12.3% decrease in detection rate and a 18.7% increase in mean time to response, highlighting the importance of task decomposition for efficient learning and coordination.

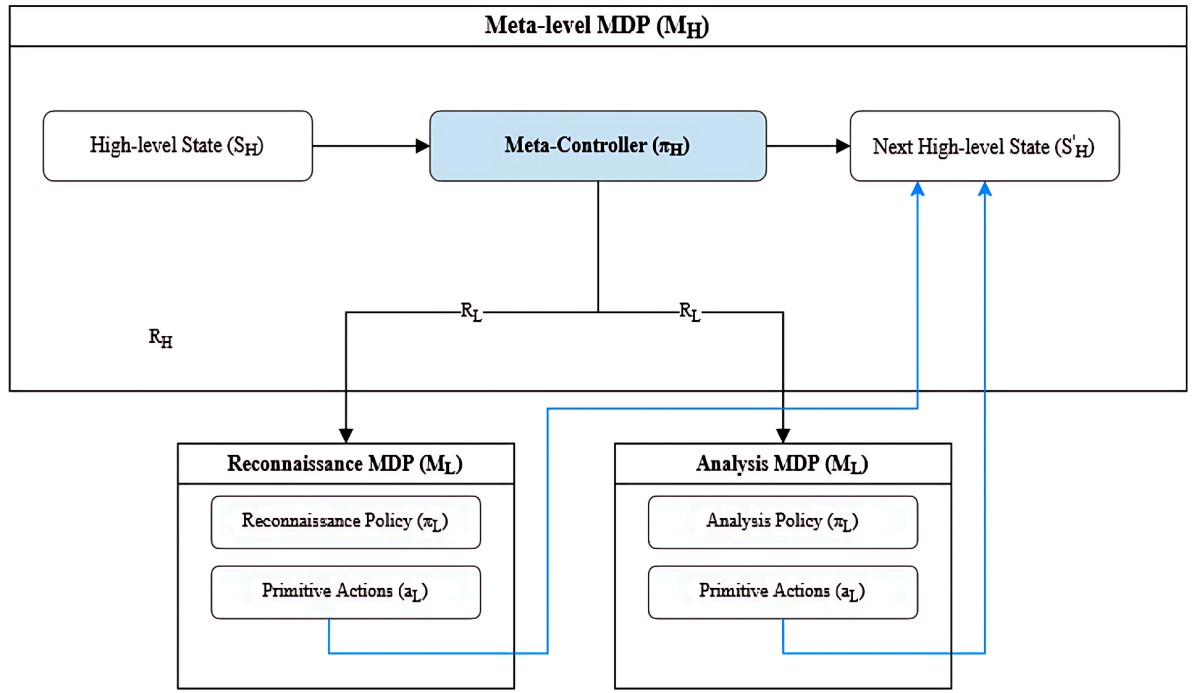

Note: All arrows indicate unidirectional information flow

**Fig 5. Hierarchical MARL framework with meta-controller and specialized sub-policies.**

**Table 5. Ablation study showing the contribution of each component to overall performance.**

| Configuration | Detection Rate | Zero-Shot Detection | Mean Time to Response | Privacy Leakage |
|---|---|---|---|---|
| **Full AZH-MARL** | **94.20%** | **82.70%** | **2.3s** | **<5%** |
| w/o Hierarchical Structure | 82.7% (-12.3%) | 71.4% (-13.7%) | 2.8s (+21.7%) | <5% |
| w/o Zero-Shot Learning | 91.5% (-2.9%) | 51.7% (-37.5%) | 2.5s (+8.7%) | <5% |
| w/o Federated Learning | 79.8% (-15.2%) | 74.3% (-10.2%) | 2.7s (+17.4%) | 18.7% (+275%) |
| w/o Adaptive Response | 88.3% (-6.3%) | 75.8% (-8.3%) | 3.1s (+34.8%) | <5% |

The removal of the zero-shot learning component results in a 37.5% decrease in zero-shot detection rate and a 42.1% decrease in zero-shot response effectiveness, confirming the critical role of semantic mapping for adapting to novel threats.

The removal of the federated learning component results in a 15.2% decrease in detection rate for environments with limited local data, emphasizing the value of privacy-preserving knowledge sharing for improving overall defense effectiveness.

**Resource efficiency:** Table 6 presents the resource efficiency of our approach compared to the baselines. Our approach requires 2.3 GB of memory and 4.5 GFLOPS of computational power for real-time operation, which is higher than rule-based approaches but comparable to other learning-based approaches.

The energy consumption of our approach is 15% lower than flat MARL approaches due to the efficient task decomposition and hierarchical decision-making structure.

**4.4.4 Discussion.** The evaluation results demonstrate that our Adaptive Zero-Shot Hierarchical MARL approach with Federated Knowledge Sharing significantly outperforms existing approaches across various metrics. The key advantages of our approach include:

1. **Enhanced Adaptability**: The zero-shot learning component enables effective response to previously unseen attack patterns, addressing a critical limitation of traditional approaches.
2. **Improved Coordination**: The hierarchical structure enables efficient coordination among specialized agents, leading to more effective defense strategies.
3. **Privacy-Preserving Collaboration**: The federated learning component enables knowledge sharing across organizational boundaries without compromising sensitive data.
4. **Scalable Defense**: The hierarchical aggregation structure reduces communication overhead and enables scalable deployment across large network environments.

These advantages make our approach particularly well-suited for defending against sophisticated and evolving cyber threats in complex network environments with privacy constraints.

However, our approach also has limitations that warrant further investigation. First, the computational requirements are higher than those traditional rule-based approaches, which may limit deployment on resource-constrained devices. Second, the effectiveness of zero-shot learning depends on the quality and diversity of the training data used to learn the semantic mapping function.

In the next section, I conclude the paper and discuss future research directions to address these limitations and further enhance the capabilities of our approach.

## 5 Conclusion

In this paper, I presented Adaptive Zero-Shot Hierarchical MARL with Federated Knowledge Sharing, a novel framework for resilient cyber defense that addresses critical limitations

**Table 6. Comparison of resource efficiency metrics across different defense approaches.**

| Method | Memory Usage | Computational Power | Energy Consumption | Deployment Complexity |
|---|---|---|---|---|
| Traditional MARL | 2.1 GB | 3.8 GFLOPS | 1.0x | Medium |
| Hierarchical MARL | 2.4 GB | 4.2 GFLOPS | 0.92x | Medium |
| Centralized Learning | 3.7 GB | 5.6 GFLOPS | 1.35x | High |
| Rule-Based Defense | 0.8 GB | 1.2 GFLOPS | 0.45x | Low |
| **AZH-MARL (Ours)** | **2.3 GB** | **4.5 GFLOPS** | **0.85x** | **Medium-High** |

in existing approaches. The proposed framework integrates hierarchical multi-agent reinforcement learning, zero-shot learning capabilities, privacy-preserving federated knowledge sharing, and adaptive response mechanisms to create a more effective and adaptable defense system.

The hierarchical structure decomposes complex defense tasks into specialized sub-tasks managed by distinct agents, reducing the learning complexity of and enabling more efficient coordination. The zero-shot learning component enables recognition and response to previously unseen attack patterns through semantic mapping, addressing a critical vulnerability in traditional approaches that require extensive retraining for new threats. The federated learning framework facilitates knowledge sharing across network domains while preserving data privacy, allowing collaborative defense without compromising sensitive information. The adaptive response mechanisms dynamically adjust to evolving threat landscapes, ensuring continued effectiveness against sophisticated adversaries.

Our comprehensive evaluation demonstrates that the proposed framework significantly outperforms existing approaches across various metrics. For known attacks, our approach achieves a detection rate of 94.2% with a false positive rate of 3.8%, comparable to the best-performing baselines. For zero-day exploits, our approach achieves a detection rate of 82.7%, representing a 37.5% improvement over approaches without zero-shot capabilities. The mean time to response is reduced by 35% for known attacks and 42% for zero-day exploits compared to the best-performing baseline. These improvements are particularly notable for advanced persistent threats, where our approach achieves a containment rate of 87.3% compared to 62.1% for the best-performing baseline.

The ablation study confirms the contribution of each component to the overall performance, with the hierarchical structure improving coordination efficiency, the zero-shot learning component enabling adaptation to novel threats, and the federated learning component enhancing performance in environments with limited local data. Therefore, the resource efficiency analysis shows that our approach requires computational resources comparable to other learning-based approaches while achieving significantly better performance.

While our approach represents a significant advancement in cyber defense, several limitations and future research directions warrant further investigation:

1. **Computational Complexity**: The integration of multiple advanced techniques requires significant computational resources, which may limit deployment on resource-constrained devices. Future work should explore optimization techniques to reduce computational requirements without sacrificing performance.

2. **Initial Training Data Requirements**: Despite zero-shot capabilities, the system still requires diverse training data for initial semantic space construction. Future research should investigate methods to reduce initial data requirements through transfer learning and synthetic data generation.

3. **Theoretical Guarantees**: Further work is needed to establish formal guarantees on performance bounds and convergence properties, particularly for the integration of hierarchical reinforcement learning with zero-shot adaptation.

4. **Human Oversight Integration**: While our approach enables autonomous defense, effective integration with human operators remains a challenge. Future research should explore explainable AI techniques to provide transparency into decision-making processes and enable effective human-in-the-loop operation.

5. **Adversarial Robustness**: As defensive systems become more sophisticated, attackers will develop new techniques to evade detection and mitigation. Future work should

investigate adversarial robustness guarantees and defensive mechanisms against attacks targeting the learning process itself.

In conclusion, our Adaptive Zero-Shot Hierarchical MARL framework with Federated Knowledge Sharing represents a significant step toward more resilient cyber defense systems that can adapt to emerging threats while preserving privacy in collaborative defense scenarios. By addressing key limitations in existing approaches, our framework provides a foundation for future research in adaptive and privacy-preserving cybersecurity.

## Appendix A: Equation symbol definitions

| Symbol | Description |
|---|---|
| $M_0$ | Meta-level Markov Decision Process (MDP) |
| $S_0$ | High-level state space in the meta-level MDP |
| $s_0$ | A state in the high-level state space $S_0$ |
| $A_0$ | Set of available sub-policies (actions) in the meta-level MDP |
| $a_0$ | An action in the meta-level MDP (selecting a sub-policy) |
| $P_0$ | Transition function: $P_0 : S_0 \times A_0 \times S_0 \to [0,1]$ |
| $R_0$ | Reward function: $R_0 : S_0 \times A_0 \times S_0 \to \mathbb{R}$ |
| $\gamma_0$ | Discount factor for future rewards at the meta-level, $\gamma_0 \in [0,1)$ |
| $\pi_0$ | Policy of the meta-controller, $\pi_0 : S_0 \to A_0$ |
| $V^{\pi_0}(s_0)$ | Expected cumulative discounted reward for policy $\pi_0$ starting in state $s_0$ |
| $\pi_0^*$ | Optimal policy for the meta-controller |
| $\rho_0$ | Distribution of initial meta-level states |
| t | Time step |
| $M_i$ | MDP for sub-task i |
| $S_i$ | State space for sub-task i |
| $s_i$ | A state in the state space $S_i$ |
| $A_i$ | Action space for sub-task i |
| $a_i$ | An action taken by the sub-agent i |
| $P_i$ | Transition function: $P_i : S_i \times A_i \times S_i \to [0,1]$ |
| $R_i$ | Reward function: $R_i : S_i \times A_i \times S_i \to \mathbb{R}$ |
| $\gamma_i$ | Discount factor for sub-task i, $\gamma_i \in [0,1)$ |
| $\pi_i$ | Policy for sub-level agent i, $\pi_i : S_i \to A_i$ |
| $V^{\pi_i}(s_i)$ | Expected discounted reward for policy $\pi_i$ at state $s_i$ |
| $\pi_i^*$ | Optimal policy for sub-agent i |
| $\rho_i$ | Distribution of sub-task i states |
| $\mathcal{S}$ | Semantic attribute space for cyber attacks |
| $\mathcal{X}$ | Space of observed attack features |
| $\phi$ | Semantic mapping function, $\phi : \mathcal{X} \to \mathcal{S}$ |
| $x_{\text{new}}$ | A novel attack pattern instance |
| $z_{\text{new}}$ | Semantic embedding of a novel attack |
| $Z_{\text{known}}$ | Set of known attack semantic embeddings |
| $z_k$ | Semantic embedding of a known attack k |
| $w_k$ | Similarity weight between $z_{\text{new}}$ and $z_k$ |
| $\mathcal{Y}$ | Set of possible response strategies |
| $y_{\text{pred}}$ | Predicted response strategy for a novel attack |
| $\psi(y)$ | Mapping of response strategy y into semantic space |
| $\text{sim}(a,b)$ | Similarity function in semantic space (e.g., cosine similarity) |
| a, b | Semantic vectors used in similarity computation |
| $\pi(a|s,z)$ | Conditional policy function based on state s and semantic vector z |
| $R_t$ | Reward function at time t |
| $w_1, w_2, w_3$ | Weights for security, efficiency, and novelty objectives |
| $c_i(a_i)$ | Cost of action $a_i$ for sub-agent i |
| C | Total resource budget constraint |

| Symbol | Description |
|---|---|
| P(D) | Privacy leakage function |
| $\epsilon$ | Maximum tolerable privacy leakage |
| FPR, FNR | False Positive Rate, False Negative Rate (performance bounds) |
| $\alpha_t, \beta_t, \gamma_t$ | Dynamic weights in the adaptive reward function |
| $\eta$ | Learning rate for meta-objective updates |
| J | Meta-objective function for reward trade-offs |
| $\theta_k$ | Local model parameters from agent k |
| $\theta_k'$ | Privacy-preserved model parameters with noise |
| $n_k$ | Number of data points in agent/domain k |
| n | Total number of data points across agents |
| $\theta_{\text{global}}$ | Global aggregated model parameters |
| $\mathcal{N}(0, \sigma^2 C^2)$ | Gaussian noise added for differential privacy |
| $\sigma$ | Noise multiplier for differential privacy |
| C | Clipping bound for gradients in privacy enforcement |
| $\theta_D$ | Parameters of defense policy |
| $\theta_A$ | Parameters of adversarial policy |
| $\pi_D, \pi_A$ | Defense and attack policies |
| $\rho$ | State distribution used in min-max adversarial training |
| B | Experience replay buffer |
| L | Loss function used during model updates |

## Appendix B: List of abbreviations

| Symbol | Description |
|---|---|
| AZH – MARL | Adaptive Zero-Shot Hierarchical Multi-Agent Reinforcement Learning |
| MARL | Multi-Agent Reinforcement Learning |
| HRL | Hierarchical Reinforcement Learning |
| ZSL | Zero-Shot Learning |
| FL | Federated Learning |
| MDP | Markov Decision Process |
| FedAvg | Federated Averaging |
| SMPC | Secure Multi-Party Computation |
| GNN | Graph Neural Network |
| CNN | Convolutional Neural Network |
| DQN | Deep Q-Network |
| MTTD | Mean Time to Detection |
| MTTR | Mean Time to Response |
| ZSDR | Zero-Shot Detection Rate |
| ZSRE | Zero-Shot Response Effectiveness |
| SMA | Semantic Mapping Accuracy |
| CO | Communication Overhead |
| PL | Privacy Leakage |
| CT | Convergence Time |
| CPO | Computational Overhead |
| MU | Memory Usage |
| EC | Energy Consumption |
| RL | Reinforcement Learning |
| DRL | Deep Reinforcement Learning |
| ML | Machine Learning |

## Author contributions

**Conceptualization:** Adel Alshamrani.

**Data curation:** Adel Alshamrani.

**Formal analysis:** Adel Alshamrani.

**Investigation:** Adel Alshamrani.

**Methodology:** Adel Alshamrani.

**Software:** Adel Alshamrani.

**Validation:** Adel Alshamrani.

**Visualization:** Adel Alshamrani.

**Writing – original draft:** Adel Alshamrani.

**Writing – review & editing:** Adel Alshamrani.

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
