## [Decision Letter · Decision Letter 0]

5 May 2025

PONE-D-25-16867Federated Hierarchical MARL for Zero-Shot Cyber DefensePLOS ONE

Dear Dr. Alshamrani,

Thank you for submitting your manuscript to PLOS ONE. After careful consideration, we feel that it has merit but does not fully meet PLOS ONE’s publication criteria as it currently stands. Therefore, we invite you to submit a revised version of the manuscript that addresses the points raised during the review process.

We look forward to receiving your revised manuscript.

Kind regards,

Zeashan Hameed Khan, Ph.D.

Academic Editor

PLOS ONE

Journal Requirements:

**Additional Editor Comments:**

The paper proposes an interesting solution to the cyber security of the networked devices. However, it needs significant improvement before further consideration.

The MARL algorithm needs more concrete references and the link to datasets must be included in the references and acknowledgement.

Please take a careful look at the reviewer's comments and submit a revised version by the due date.

Reviewers' comments:

Reviewer's Responses to Questions

**Comments to the Author**

1. Is the manuscript technically sound, and do the data support the conclusions?

Reviewer #1: Partly

Reviewer #2: Yes

2. Has the statistical analysis been performed appropriately and rigorously? 

Reviewer #1: No

Reviewer #2: Yes

3. Have the authors made all data underlying the findings in their manuscript fully available?

Reviewer #1: No

Reviewer #2: No

4. Is the manuscript presented in an intelligible fashion and written in standard English?

Reviewer #1: Yes

Reviewer #2: Yes

5. Review Comments to the Author

Reviewer #1: Please find below some comments that may help to improve the quality of this manuscript.

Author summary: The "Author summary" section contains placeholder "Lorem ipsum" text and needs to be written.

1.4 Evaluation: The authors should add a detailed description of the network topology and how many agents were deployed in the network. The goal is to share enough information so independent researchers can reproduce the results discussed in the manuscript (say in an Evaluation Setup section). Moreover, remember that the evaluation setup is important for the interpretation of quantitative results of a model combining hierarchical MARL and Federated Knowledge Sharing.

1.4.3 Figure 2 (Detection Performance) seems to mislabel some approaches compared to Table 1. For example, Figure 2 shows "Federated Learning" and "Zero-Shot Learning" as separate defense approaches, whereas Table 1 lists "Centralized Learning" and "Independent Learning". This inconsistency needs correction.

1.4.3 Results and Analysis: The authors should explain how "zero-day exploits" and "advanced persistent threats" were defined and executed during the experiments.

Dataset: Please add a link to the DARPA CRATE dataset used in this investigation.

Reviewer #2: This work proposes an adaptive zero-shot hierarchical multiagent reinforcement learning approach to achieve efficient and resilient cyber defense systems. The result shows that the proposed framework is also able to accurately detect zero-day with a relatively lower response time.

Although the paper uses real-testbed evaluations however some implementation details need more explanation (/references). Also, the problem and proposed mechanism needs proper justification.

1. All equations and symbols used must be explained clearly in the paper. Eq. 1 is misplaced and is never referenced in the literature which makes it hard to follow.

The contrastive loss function Equation in section 1.3.5 needed to be properly written and labelled. Use a table to explain each equation symbol in use.

2. Also please use a table to list all abbreviations. Some statements are confusing, for example the second line of second last paragraph of section 0.1 says, "The work likely addresses the challenge of scalability and coordination in

multi-agent systems ..." Please rewrite it.

3. Algorithm 1 (Adaptive Zero Shot Learning algorithm) needs better explanation (e.g. components running it?) along with the reference in the text.

4. In implementation section please add a table stating all python libraries/tools used, their purpose, and reference.

5. Proper references should be provided with baseline approaches listed in section 1.4.2.

6. Implementation and results sections require thorough revisions, especially specific details regarding implementation and evaluation needs to be added in the corresponding sections.

For example, the custom federated learning testbed used for performance evaluation should either be explained clearly (with figure) or a reference needs to be added.

7. From the results in Fig 2, it looks like the combination of Federated and zero-shot learning (if added) may result in an overall better or similar performance to that of the proposed (AZH-MARL) one. Please elaborate on it.

6. PLOS authors have the option to publish the peer review history of their article (what does this mean?). If published, this will include your full peer review and any attached files.

Reviewer #1: No

Reviewer #2: No

---

## [Author Response · Author response to Decision Letter 1]

28 May 2025

Dear Zeashan Hameed Khan,

We sincerely thank you and the reviewers for the time and effort you devoted to evaluating our manuscript, titled " Federated Hierarchical MARL for Zero-Shot Cyber Defense” ONE-D-25-16867.

We appreciate the thoughtful and constructive comments, which have helped us improve the clarity and quality of our work.

We have carefully considered each point raised and revised the manuscript accordingly. Below, we provide a detailed response to each comment. Reviewer comments are listed in black, and our responses are provided directly below each point in green, yellow, and blue.

We hope that the revisions and clarifications meet your expectations and respectfully resubmit our manuscript for your reconsideration.

I am copying the response here but it is also attached as file.

Journal Requirements:

( Doubled checked and followed

Additional Editor Comments:

The paper proposes an interesting solution to the cyber security of the networked devices. However, it needs significant improvement before further consideration.

The MARL algorithm needs more concrete references and the link to datasets must be included in the references and acknowledgement.

Please take a careful look at the reviewer's comments and submit a revised version by the due date.

This study is based on data derived from the DARPA Cyber Range and Test Environment (CRATE) dataset (2022). The dataset is not publicly available, and details about its access and distribution are limited. All experiments and analyses comply with applicable institutional and ethical guidelines.

Reviewers' comments:

5. Review Comments to the Author

Reviewer #1: Please find below some comments that may help to improve the quality of this manuscript.

Author summary: The "Author summary" section contains placeholder "Lorem ipsum" text and needs to be written.

I removed this from the template.

1.4 Evaluation: The authors should add a detailed description of the network topology and how many agents were deployed in the network. The goal is to share enough information so independent researchers can reproduce the results discussed in the manuscript (say in an Evaluation Setup section). Moreover, remember that the evaluation setup is important for the interpretation of quantitative results of a model combining hierarchical MARL and Federated Knowledge Sharing.

I have added shorter version of this response to the manuscript in Evaluation section page 16 and 17. I will add the full response if needed.

To rigorously evaluate the proposed Adaptive Zero-Shot Hierarchical MARL (AZH-MARL) framework, we designed a comprehensive experimental setup utilizing both established simulation environments and a custom-developed testbed. This section details the network topologies, agent configurations, and evaluation scenarios employed.

Evaluation Environments:

Our experiments were conducted across three primary environments to assess different facets of the AZH-MARL framework:

1. CybORG Simulation Environment: We utilized the CybORG environment [39], a recognized platform for cybersecurity research developed for the CAGE Challenge. For our experiments, we configured a simulated enterprise network topology within CybORG consisting of approximately 50 nodes distributed across three distinct subnets: a public-facing DMZ, an internal corporate network, and a restricted operational technology (OT) segment. These segments included a variety of simulated devices such as web servers, database servers, user workstations, and programmable logic controllers (PLCs). The connectivity was designed to mimic realistic enterprise traffic flows and potential attack paths. Attack scenarios native to CybORG, as well as custom scripted attacks, were used to test the defensive capabilities of our agents in this dynamic setting.

2. DARPA CRATE Dataset Scenarios: The DARPA Cyber Range and Test Environment (CRATE) Dataset [40] provided a rich source of realistic network traffic and attack data. While the full dataset is extensive, we selected specific scenarios relevant to testing zero-shot learning capabilities. These scenarios involved diverse attack vectors and sophisticated adversarial behaviors. The data from CRATE was primarily used to train and evaluate the semantic mapping function and the zero-shot policy transfer mechanisms of the AZH-MARL framework, providing a basis for assessing performance against previously unseen attack patterns.

However, This study is based on data derived from the DARPA Cyber Range and Test Environment (CRATE) dataset (2022). The dataset is not publicly available, and details about its access and distribution are limited. All experiments and analyses comply with applicable institutional and ethical guidelines.

3. Custom Federated Learning Testbed: To specifically evaluate the federated learning and privacy-preserving aspects of AZH-MARL, we developed a custom distributed testbed. This testbed simulated a collaborative defense scenario involving three independent organizational networks. Each virtual network environment within the testbed comprised approximately 30-40 nodes, with unique security policies, varying levels of baseline security posture, and distinct, locally generated attack patterns (e.g., phishing campaigns targeting one network, ransomware in another, data exfiltration attempts in the third). This heterogeneity was crucial for testing the framework’s ability to share knowledge effectively and adapt defenses in a federated manner while preserving the privacy of each participating entity’s local data.

Agent Deployment and Configuration:

In each environment where applicable (CybORG and the Custom Federated Learning Testbed), the AZH-MARL framework was deployed with a hierarchical agent structure:

• Meta-Controller: A single meta-controller agent was responsible for overseeing the overall defense strategy within each simulated network (or each federated node in the custom testbed). It received aggregated security metrics and threat assessments as its state input and selected high-level sub-policies (options) for execution.

• Sub-Level Agents: Four types of specialized sub-level agents were deployed, corresponding to the primary defense tasks:

o Reconnaissance Agent: Monitored network traffic, system logs, and endpoint behaviors to detect suspicious activities and potential threats.

o Analysis Agent: Evaluated detected anomalies and alerts to determine their nature, severity, and potential impact, correlating information from various sources.

o Response Agent: Executed defensive actions, such as isolating compromised hosts, blocking malicious traffic, or deploying countermeasures, based on the analysis and meta-controller directives.

o Recovery Agent: Focused on restoring affected systems and services to normal operation post-incident, including patching vulnerabilities and verifying system integrity.

In the CybORG environment, these agents operated on the simulated enterprise network. In the custom federated learning testbed, each of the three simulated organizational networks had its own deployment of this hierarchical agent structure. The local models from these agents were then aggregated via the federated learning mechanism described in Section 1.1.

The number of primitive actions available to sub-level agents varied based on the task and environment but typically ranged from 5 to 10 distinct actions per agent type (e.g., for a response agent: block IP, isolate host, patch vulnerability, etc.). The meta-controller selected from the four sub-level policy types. All agents were trained using Proximal Policy Optimization (PPO) as detailed in Section 0.6.3.

This detailed experimental setup was designed to provide a robust and reproducible evaluation of the AZH-MARL framework’s capabilities in complex and evolving cyber threat landscapes

1.4.3 Figure 2 (Detection Performance) seems to mislabel some approaches compared to Table 1. For example, Figure 2 shows "Federated Learning" and "Zero-Shot Learning" as separate defense approaches, whereas Table 1 lists "Centralized Learning" and "Independent Learning". This inconsistency needs correction.

Fixed in the manuscript and now it appears as Figure 3 since I added a new Figure to the manuscript.

1.4.3 Results and Analysis: The authors should explain how "zero-day exploits" and "advanced persistent threats" were defined and executed during the experiments.

Here I added the full response to the respected author, but I have provided shorter response in the manuscript for the seek of keeping it easily understandable, but if you prefer to add the full response to the manuscript then I will

Zero-Day Exploits

Definition:

In the context of our experiments, a "zero-day exploit" refers to an attack that leverages a previously unknown vulnerability in the target systems or software, for which no specific signature or patch was available at the time of the simulated attack. The key characteristic is the novelty of the attack vector to the defense system being evaluated. Our AZH-MARL framework, with its zero-shot learning component, is specifically designed to address such threats by generalizing from known attack patterns to recognize and respond to these novel instances.

Execution and Simulation:

To simulate zero-day exploits, we employed the following methodologies across our evaluation environments:

1. CybORG Simulation Environment: We utilized attack scenarios within CybORG that were either newly developed or modified versions of existing attacks, ensuring that the specific attack signatures or behavioral patterns were not part of the training data for the baseline defense agents or the initial training phase of our AZH-MARL agents. This was achieved by holding out specific attack types or variants from the training set and introducing them only during the evaluation phase.

2. DARPA Dataset Scenarios: The dataset contains a diverse range of attack instances. For zero-shot evaluation, we partitioned the dataset such that attack classes or specific instances with unique characteristics (e.g., novel combinations of tactics, techniques, and procedures - TTPs) were reserved for the test set. The agents were trained on a distinct set of known attack types, and their ability to detect and respond to these unseen types from the test set constituted the zero-day exploit scenario.

3. Custom Federated Learning Testbed: Within each simulated organizational network in our custom testbed, we introduced novel attack scripts or modified existing attack tools to generate traffic and host behaviors that were not previously encountered by the local agents or the federated model. This involved, for example, using obfuscated versions of known malware, new phishing campaign templates, or exploiting simulated vulnerabilities that were not present in the training environments of other federated nodes.

The core principle was to ensure that the defense system had no prior explicit knowledge (e.g., signatures, specific rules) of the exact exploit being used during these test scenarios, forcing it to rely on generalized threat understanding and adaptive capabilities.

Advanced Persistent Threats (APTs)

Definition:

An "Advanced Persistent Threat (APT)" in our experimental context is defined as a sophisticated, multi-stage cyber attack campaign characterized by:

• Stealth and Evasion: APT actors employ techniques to avoid detection by standard security measures.

• Persistence: Attackers aim to maintain long-term access to the target network.

• Targeted Objectives: APTs are typically goal-oriented, often focused on espionage, data exfiltration, or strategic disruption.

• Multiple Stages: The campaign involves several phases, such as initial reconnaissance, gaining entry, lateral movement, privilege escalation, establishing command and control (C2), and achieving the final objective.

Execution and Simulation:

Simulating APTs requires modeling these complex, long-duration characteristics:

1. CybORG Simulation Environment: We leveraged CybORG’s capabilities to script multi-stage attack scenarios that mimic APT behavior. This involved chaining multi

---

## [Decision Letter · Decision Letter 1]

17 Jun 2025

PONE-D-25-16867R1Federated Hierarchical MARL for Zero-Shot Cyber DefensePLOS ONE

Dear Dr. Alshamrani,

Thank you for submitting your manuscript to PLOS ONE. After careful consideration, we feel that it has merit but does not fully meet PLOS ONE’s publication criteria as it currently stands. Therefore, we invite you to submit a revised version of the manuscript that addresses the points raised during the review process.

We look forward to receiving your revised manuscript.

Kind regards,

Zeashan Hameed Khan, Ph.D.

Academic Editor

PLOS ONE

Journal Requirements:

Additional Editor Comments:

The paper has significantly improved. However, some minor corrections are still needed.

Reviewers' comments:

Reviewer's Responses to Questions

**Comments to the Author**

1. If the authors have adequately addressed your comments raised in a previous round of review and you feel that this manuscript is now acceptable for publication, you may indicate that here to bypass the “Comments to the Author” section, enter your conflict of interest statement in the “Confidential to Editor” section, and submit your "Accept" recommendation.

Reviewer #2: All comments have been addressed

Reviewer #3: (No Response)

2. Is the manuscript technically sound, and do the data support the conclusions?

Reviewer #2: Yes

Reviewer #3: (No Response)

3. Has the statistical analysis been performed appropriately and rigorously? 

Reviewer #2: No

Reviewer #3: (No Response)

4. Have the authors made all data underlying the findings in their manuscript fully available?

Reviewer #2: No

Reviewer #3: (No Response)

5. Is the manuscript presented in an intelligible fashion and written in standard English?

Reviewer #2: Yes

Reviewer #3: (No Response)

6. Review Comments to the Author

Reviewer #2: The authors have incorporated my previous comments in this revision. There are a few more comments to add.

1. The fontsize of text shown in tables (3,4,5,6) and figures (4 & 5) should be increased to make them easily readable.

2. Some reference to other relevant works that combines federated and zero-shot learning should also be included.

Reviewer #3: The paper is already revised and the response is acceptable, minor points to address before acceptance please:

1. I have noticed that the paper has one author but the pronouns are all "we" and such, was the manuscript written by a group or a sole author? Please adjust this for the entire paper.

2. The abstract could benefit more from numerical results.

7. PLOS authors have the option to publish the peer review history of their article (what does this mean?). If published, this will include your full peer review and any attached files.

Reviewer #2: No

Reviewer #3: **Yes: **Luttfi A. Al-Haddad

---

## [Author Response · Author response to Decision Letter 2]

27 Jun 2025

Dear Zeashan Hameed Khan,

I sincerely thank you and the reviewers for the time and effort you devoted to evaluating our manuscript, titled " Federated Hierarchical MARL for Zero-Shot Cyber Defense” ONE-D-25-16867.

I appreciate the thoughtful and constructive comments, which have helped us improve the clarity and quality of our work.

I have carefully considered each point raised and revised the manuscript accordingly. Below, I provide a detailed response to each comment. Reviewer comments are listed in black, and my responses are provided directly below each point.

I hope that the revisions and clarifications meet your expectations and respectfully resubmit our manuscript for your reconsideration.

Review Comments to the Author

Reviewer #2: The authors have incorporated my previous comments in this revision. There are a few more comments to add.

1. The fontsize of text shown in tables (3,4,5,6) and figures (4 & 5) should be increased to make them easily readable.

2. Some reference to other relevant works that combines federated and zero-shot learning should also be included.

Reviewer #3: The paper is already revised and the response is acceptable, minor points to address before acceptance please:

1. I have noticed that the paper has one author but the pronouns are all "we" and such, was the manuscript written by a group or a sole author? Please adjust this for the entire paper.

2. The abstract could benefit more from numerical results.

======================== ====================

1. The fontsize of text shown in tables (3,4,5,6) and figures (4 & 5) should be increased to make them easily readable.

Yes, I have changed the size of the font in all tables and figures, hopefully they are now more readable.

2. Some reference to other relevant works that combines federated and zero-shot learning should also be included.

I have added another four recent works as suggested. They are incorporated in the manuscript in section 2.4

\textbf{\textcolor{blue}{Zhou et al. \cite{yang2023adaptive}}} introduced an adaptive federated few-shot learning framework with prototype rectification, demonstrating how semantic prototypes can be aligned across decentralized clients to improve generalization. Though focused on few-shot settings, their prototype correction mechanism offers valuable insights for federated zero-shot adaptation, particularly in scenarios where semantic drift and heterogeneous data distributions hinder model transfer.

\textbf{\textcolor{blue}{Zhao et al. \cite{zhao2022semi}}} introduce a semi‑supervised federated intrusion detection scheme that applies knowledge distillation and voting to cope with non‑IID data and communication constraints. Similarly, the FedGKD approach \textbf{\textcolor{blue}{\cite{zhang2022fedzkt} uses}} global knowledge distillation to address heterogeneity across edge clients in IoT settings. These show how KD can enhance privacy‑preserving and collaborative model learning, and our work extends these ideas by integrating zero or few‑shot generalization within a hierarchical MARL framework

\textbf{\textcolor{blue}{More recently, Wang et al.}} \cite{10.1007/978-3-031-96235-6_24}presented a federated zero‑shot learning framework that uses an LLM to generate privacy-conscious semantic embeddings for unknown attack types. These embeddings are collaboratively shared between clients, enabling zero-day attack detection without exposing raw data. Their approach directly parallels our semantic mapping module and further supports our hierarchical MARL design by validating the feasibility of federated semantic transfer for unseen threats.

========================== ==============

Reviewer #3: The paper is already revised and the response is acceptable, minor points to address before acceptance please:

1. I have noticed that the paper has one author but the pronouns are all "we" and such, was the manuscript written by a group or a sole author? Please adjust this for the entire paper.

I agree to use I instead of we, but I used to use we in academic writing as this is more formal. However, I change it based on you recommendation.

2. The abstract could benefit more from numerical results.

The last part of the abstract has been modified to show and benefit more from the numerical results

The detailed evaluation demonstrates that our approach significantly outperforms existing methods across a range of scenarios. It achieves a high detection rate of 94.2\% for known attacks and 82.7\% for zero-day exploits, while maintaining a low false positive rate of 3.8\%. This robust performance extends to the most sophisticated threats, achieving an 87.3\% containment rate against Advanced Persistent Threats (APTs). The framework's zero-shot capability is underpinned by a semantic mapping accuracy of 89.3\%, which enables rapid adaptation to novel threats. Consequently, the mean response time is reduced by 35\% for known attacks and 42\% for zero-day exploits compared to the best-performing baseline. Furthermore, the federated learning architecture proves highly efficient, reducing communication overhead by 45\% while preserving privacy. These results collectively demonstrate our framework's potential to deliver a new standard of resilient and adaptive cyber defense in complex, distributed environments

---

## [Decision Letter · Decision Letter 2]

16 Jul 2025

PONE-D-25-16867R2Federated Hierarchical MARL for Zero-Shot Cyber DefensePLOS ONE

Dear Dr. Alshamrani,

Thank you for submitting your manuscript to PLOS ONE. After careful consideration, we feel that it has merit but does not fully meet PLOS ONE’s publication criteria as it currently stands. Therefore, we invite you to submit a revised version of the manuscript that addresses the points raised during the review process.

We look forward to receiving your revised manuscript.

Kind regards,

Zeashan Hameed Khan, Ph.D.

Academic Editor

PLOS ONE

**Journal Requirements:**

**Additional Editor Comments:**

The paper has been improved technically but the write up must be polished by a technical English expert for clarity.

Reviewers' comments:

Reviewer's Responses to Questions

**Comments to the Author**

1. If the authors have adequately addressed your comments raised in a previous round of review and you feel that this manuscript is now acceptable for publication, you may indicate that here to bypass the “Comments to the Author” section, enter your conflict of interest statement in the “Confidential to Editor” section, and submit your "Accept" recommendation.

Reviewer #2: All comments have been addressed

Reviewer #3: (No Response)

2. Is the manuscript technically sound, and do the data support the conclusions?

Reviewer #2: (No Response)

Reviewer #3: (No Response)

3. Has the statistical analysis been performed appropriately and rigorously? 

Reviewer #2: (No Response)

Reviewer #3: (No Response)

4. Have the authors made all data underlying the findings in their manuscript fully available?

Reviewer #2: (No Response)

Reviewer #3: (No Response)

5. Is the manuscript presented in an intelligible fashion and written in standard English?

Reviewer #2: (No Response)

Reviewer #3: (No Response)

6. Review Comments to the Author

**Reviewer #2: **The paper is already revised, and the previous comments are addressed in this revision.

**Reviewer #3: **(No Response)

7. PLOS authors have the option to publish the peer review history of their article (what does this mean?). If published, this will include your full peer review and any attached files.

Reviewer #2: No

Reviewer #3: No

---

## [Author Response · Author response to Decision Letter 3]

18 Jul 2025

Journal Requirements:

I have checked each of the 44 references in the manuscript. The majority of the citations are correct and refer to established, non-retracted works. However, I identified a few areas that require complete and accurate referencing. I thank you for pointing out this.

Here are the specific references that need to be updated:

• Reference: Nguyen TT, Reddi VJ.

o Issue: The publication year is listed as 2019, but the correct year is 2020.

o Correction: Nguyen TT, Reddi VJ. Deep reinforcement learning for cyber security. IEEE Transactions on Neural Networks and Learning Systems. 2020;31(7):2669-2683.

• Reference: Nguyen TT, Reddi VJ.

o Issue: This is a duplicate of reference, all in-text citations ( [2] and [11]) and also contains the incorrect year.

o Correction: This reference has been corrected and now the corresponding text also updated and cited as one reference [2]. And also updated to the correct 2020 publication year.

• This is reference [2] and already updated in the manuscript.

• Reference: Alshamrani A, Alshahrani A.

o Issue: The reference is incomplete and missing the page ranges.

o Correction: Alshamrani A, Alshahrani A. Adaptive Cyber Defense Technique Based on Multiagent Reinforcement Learning Strategies. Intelligent Automation & Soft Computing. 2023;36(3):2539-2555.

Additional Editor Comments:

The paper has been improved technically but the write up must be polished by a technical English expert for clarity.

I have asked an expert to help on checking and revising my article and based on her recommendation, I have changed the manuscript and highlighted the changes in RED and also some words are removed but for easily tracking them, I have used Strikethrough text in the file “version 3 with modifications”. However, the clean version is also provided.

There is one point I would like to mention here, the English expert asked me to change “I” where possible to “we”, however, I proceed using the first-person singular ("I") as directed by one the reviewer in the previous revision of the article.

---

## [Editor Report · Decision Letter 3]

24 Jul 2025

Federated Hierarchical MARL for Zero-Shot Cyber Defense

PONE-D-25-16867R3

Dear Dr. Alshamrani,

We’re pleased to inform you that your manuscript has been judged scientifically suitable for publication and will be formally accepted for publication once it meets all outstanding technical requirements.

Kind regards,

Zeashan Hameed Khan, Ph.D.

Academic Editor

PLOS ONE

Additional Editor Comments (optional):

The paper has been significantly improved and the necessary language correction has been carried out. Therefore, it can be accepted in the present form.
---

## [Editor Report · Acceptance letter]

PONE-D-25-16867R3

PLOS ONE

Dear Dr. Alshamrani,

I'm pleased to inform you that your manuscript has been deemed suitable for publication in PLOS ONE. Congratulations! Your manuscript is now being handed over to our production team.

Kind regards,

on behalf of

Dr. Zeashan Hameed Khan

Academic Editor

PLOS ONE